# Diagnostic utility of quantitative analysis of microRNA in bile samples obtained during endoscopic retrograde cholangiopancreatography for malignant biliary strictures

**Noriyuki Kuniyoshi[1], Hiroo Imazu[2]\*, Ryota Masuzaki[1], Motomi Yamazaki[1], Suguru Hamana[1], Shuzo Nomura[1], Jo Hayama[1], Rota Osawa[2], Koji Yamada[2], Mariko Fujisawa[1], Kei Saito[1], Hirofumi Kogure[1]**

1 Division of Gastroenterology and Hepatology, Department of Medicine, Nihon University School of Medicine, Itabashi-ku, Tokyo, Japan, 2 Division of Gastroenterology and Hepatology, Department of Medicine, Nihon University School of Medicine, Chiyoda-ku, Tokyo, Japan

\* imazu.hiroo@nihon-u.ac.jp

## Abstract

### Background

The sensitivity of bile cytology for malignant biliary strictures is not adequate. To overcome this limitation, we evaluated whether quantitative analysis of microRNAs (miRNAs) in bile can provide a precise diagnosis of malignant biliary strictures due to pancreatic cancer (PC) and biliary tract cancer (BTC).

### Methods

This was a retrospective evaluation of miRNA levels in stored bile samples of patients with PC, BTC or benign biliary stricture obtained during biliary drainage from April 2019 to December 2021 at our institution. A total of 113 patients (PC; n = 40, BTC; n = 38, control; n = 35) were enrolled. The miRNA candidates to be quantified were determined with microarray analysis from each 3 patients with PC, BTC and controls.

### Results

Using microarray analysis, we confirmed four significantly up-regulated miRNAs (miR-1275, miR-6891-5p, miR-7107-5p, miR-3197) in patients with PC and BTC compared to control patients. Quantitative PCR was then performed in 113 bile samples for these miRNAs. miR-1275 was significantly upregulated in PC (p = 0.003) and BTC (p = 0.049) compared to controls, miR-6891-5p was significantly upregulated in PC compared to controls (p = 0.025). In particular, a combination of bile cytology and miR-1275 in bile showed a sensitivity of 77.5% (95% CI, 70.7–77.5%), specificity of 100% (95% CI, 92.2–100%) and an area under the curve (AUC) of 0.93, and provided a significantly greater additional diagnostic effect than bile cytology alone (p = 0.014).

**Data Availability Statement:** All relevant data are within the paper and its Supporting Information files.

**Funding:** No funding was received for this research or the publication of this article.

**Competing interests:** The authors declare that there is no conflict of interest regarding the publication of this article.

## Conclusions

This study suggest that bile miRNAs could be potential biomarkers for pancreato-biliary diseases, particularly miR-1275 and miR-6891-5p may be helpful in the diagnosis of PC and BTC.

## Introduction

Pancreatic cancer (PC) and biliary tract cancer (BTC) are the major causes of malignant biliary stricture. However, it is often challenging to distinguish between benign and malignant biliary stricture by imaging tests alone. Bile samples can be endoscopically obtained during endoscopic retrograde cholangiopancreatography (ERCP) and used for cytological diagnosis. However, the sensitivity of bile cytology for malignant biliary stricture is reported to be as low as 6–32% [1]. Moreover, other endoscopic techniques that provide pathological or cytological diagnosis of malignant biliary strictures, such as tissue sampling with ERCP, EUS-guided fine needle aspiration (EUS-FNA) and peroral cholangioscopy (POCS), have been reported. However, the diagnostic accuracies of these methods for malignant biliary stricture also remain unsatisfactory [2–4]. In addition, the sensitivity and specificity of standard serum markers of PC and BTC, such as CEA and CA19-9, are also not enough to provide a differential diagnosis between benign and malignant biliary stricture [5, 6].

Recently, microRNAs (miRNAs) have been applied as cancer biomarkers due to their biological stability and close association with carcinogenesis [7]. miRNAs are short noncoding RNAs consisting of 18–25 nucleotides that function by targeting specific mRNA moieties for translational repression or degradation, thereby regulating several biological processes, including cell proliferation, migration, invasion, survival, and metastasis [8, 9]. There are few reports of utilizing bile samples for miRNA-based diagnosis of PC and BTC thus far, and various reagents have been used for miRNA isolation [10–14]. Thus, the utility of evaluating specific miRNAs in bile samples in the diagnosis of PC and BTC is still unknown. The quantification of the miRNAs in bile may overcome the diagnostic limitation observed in the conventional histocytological diagnosis with ERCP, EUS-FNA and peroral POCS.

The aim of the present study was to evaluate whether quantitative analysis of the selected miRNAs in bile combined with bile cytology can provide a precise diagnosis of PC and BTC.

## Material and methods

### Study design

In our institution, bile samples are collected for cytohistological and/or bacteriological examination in all patients with biliary stricture during ERCP, and the residual bile is routinely centrifuged and stored at -80˚C for further molecular biological examination. This was a retrospective evaluation of miRNAs in stored bile samples of patients with PC, BTC or benign biliary stricture who were obtained during diagnostic and/or therapeutic ERCP from April 2019 to December 2021 at Nihon University Itabashi Hospital, Tokyo, Japan. Inclusion criteria were as follows: (1) patients in whom bile samples were collected during ERCP for suspicious malignant biliary stricture; (2) histological diagnosis of PC or BTC was obtained with endoscopic cytohistology or surgical histology. As a benign control to evaluate the utility of specific miRNAs to detect PC or BTC, consecutive patients with benign biliary stricture consisting of choledocholithiasis, autoimmune pancreatitis (AIP) and chronic pancreatitis (CP) during the

study period were also enrolled, according to the number of patients with PC or BTC. In cases of benign biliary stricture, the final diagnosis was based on negative results of endoscopic cyto-histology and the consensus of the patient's clinical course and multimodal imaging tests, including computed tomography, endosonography and/or magnetic resonance imaging. In addition, this study was conducted in two steps as follows: (1) the determination of specific miRNAs in bile that can detect PC or BTC and (2) the evaluation of the diagnostic value of quantitative analysis of selected miRNAs combined with bile cytology to diagnose PC and BTC. Patients who had already taken chemotherapy or radiotherapy for PC, BTC or other malignant diseases were excluded from the study. The study protocol conformed to the ethical guidelines of the 1975 Helsinki Declaration and was approved by the Ethics Committee of Nihon University School of Medicine (RK-200114-5). All patients provided written informed consent for the collection of bile during ERCP for further molecular biological examination prior to ERCP. We accessed patient data for the first time after June 2020, when we received approval from the Ethics Committee.

## Sampling and storage of bile

All bile samples were collected during diagnostic and/or therapeutic ERCP. After biliary can-nulation, a small amount of contrast was injected into the bile duct to confirm biliary stricture. Then, the catheter was advanced above the biliary stricture, and bile samples were aspirated into the catheter and collected for cytohistological and/or bacteriological examination. More-over, approximately 1–5 ml of residual bile sample was centrifuged at 10,000×g for 10 minutes at 4˚C and stored at -80˚C for further molecular biological examination.

## miRNA isolation and quantification

Total cellular RNA was isolated from 1.0mL of each bile sample using QIAzol Lysis Reagent (Qiagen) and purified using the miRNeasy Serum/Plasma kit (Qiagen) according to the manu-facturer's protocol. To elute the miRNA-enriched fraction and remove large RNA (>200 nt) from the total RNA, we added 70% ethanol to the aqueous phase and flow-through the RNeasy MinElute spin column following protein precipitation by chloroform extraction. Complemen-tary DNA was generated using 8 ng of RNA in conjunction with specific miRNA reverse tran-scription primers and a miRNA reverse transcription kit (Qiagen) according to the manufacturer's protocol with a GeneAmp PCR System 9700 (Applied Biosystems). Triplicate samples of the resultant cDNA were subjected to SYBR-Green-based real-time PCR using miRCURY LNA SYBR Green PCR Kits (Qiagen) according to the manufacturer's protocol with a 7500 Real-Time PCR System (Applied Biosystems). Expression levels for all candidate miRNAs were normalized to miR-16, which is frequently used as a normalizer because of its high expression and relatively invariant level across various samples [15, 16]. For quantifica-tion, the relative Ct method (2-ΔΔCt method) was applied. The Ct value of miR-16 was sub-tracted from the Ct value of the target miRNA in each sample, which gave the ΔCt. The mean ΔCt of the control samples was then subtracted from the ΔCt of the tested samples, which gave the ΔΔCt. The fold change of miRNA was calculated by the Equation 2−ΔΔCt.

## miRNA expression analysis by microarray

To identify bile miRNAs that are aberrantly expressed in PC and BTC, miRNA microarray analysis, including 2578 human mature miRNAs from RNA pools from each 3 patients with PC, BTC and controls, was performed with a GeneChip™ miRNA 4.0 Array (Thermo Fisher Scientific). Screened patients were matched by sex, age and laboratory values. A total of 130 ng of total RNA from each of the 9 samples was labeled using a FlashTag™ Biotin HSR RNA

Labeling Kit (Thermo Fisher Scientific) and hybridized to a GeneChip™ miRNA 4.0 Array (Thermo Fisher Scientific) according to the manufacturer's instructions. All hybridized microarrays were scanned by a GeneChip scanner (Thermo Fisher Scientific). Relative hybridization intensities and background hybridization values were calculated using Expression Console™ (Thermo Fisher Scientific). We processed the raw CEL files for gene-level analysis with median polish summarization and quantile normalization by Transcriptome Analysis Console Software™ (Thermo Fisher Scientific) and obtained normalized intensity values.

To identify up- or down-regulated genes, we calculated ratios (nonlog scaled fold-change) from the normalized intensities of each gene for comparisons between control and experimental samples. Then, we established criteria for regulated genes: (up-regulated genes) ratio $\geq 2.0$, (downregulated genes) ratio $\leq 0.5$. Based on the strength of regulation and high abundance in the microarray analysis, several miRNAs were chosen for further validation.

## Statistical analysis

In this study, the miRNA expression profiles of the following three paired clinical conditions were statistically compared: i) PC versus control, ii) BTC versus control and iii) PC versus BTC. Sample characteristics, laboratory characteristics, and miRNA expression levels of patients in the PC, BTC, and control groups were compared using analysis of variance/Welch's test for normally distributed continuous variables, nonparametric Kruskal–Wallis tests/ Mann–Whitney's U test for nonnormally distributed continuous variables, and Pearson's $\chi2$ tests ($\chi2$-test or Fisher's exact test) for categorical variables. All data were tested for normality using the Shapiro–Wilk test. For correlations, Spearman's rank correlation was used. Receiver operating characteristic (ROC) curves and the area under the curve (AUC) were used to assess the diagnostic performance of the quantitative analysis of different miRNAs alone and in combination with ERCP cytology. The cutoff value was determined by ROC curve, which revealed the highest sum of sensitivity and specificity for the diagnosis (Youden index) [17]. P values < 0.05 were considered statistically significant. The software used was GraphPad Prism 9 (GraphPad Software, San Diego, California) and R version 2.6.2 (R Foundation for Statistical Computing, http://www.R-project.org).

## Results

### Study population and characteristics

Forty patients with PC, 38 with BTC and 35 controls were enrolled in this study, and bile samples were collected and analyzed. The patient characteristics are described in Table 1. The rate of females was significantly higher in the PC cohort than in the BTC cohort. Smoking status was more prevalent among BTC patients but similar between PC and control patients. For the patients with PC and BTC, 35% of patients (27/78) were clinical stage IV according to the Union for International Cancer Control (UICC) TNM staging at the time of bile sampling. Laboratory characteristics are described in Table 2. The median serum levels of CA19-9 and total bilirubin were significantly higher in the PC and BTC groups than in the control. The CRP level was significantly higher in the BTC group than in the control.

### Determination of specific miRNAs for the detection of PC or BTC by microarray analysis

Array-based screening of 2578 miRNAs from RNA pools of bile samples of 9 patients, consisting of 3 patients with PC, 3 with BTC and 3 with control, identified several deregulated miRNAs in PC and BTC compared to control patients. In relative quantification, 35 of 2578

**Table 1. Patient characteristics.**

| Variable | PC (n = 40) | BTC (n = 38) | Control (n = 35) | P value | PC vs Control | BTC vs Control | PC vs BTC |
|---|---|---|---|---|---|---|---|
| Age,mean(s.d.) | 74.8(8.8) | 73.9(10.4) | 70.5(12.9) | 0.185 | | | |
| Female sex,n(%) | 24(60%) | 12(32.0%) | 14(40%) | 0.034 | 0.084 | 0.453 | 0.012 |
| Smoking status, n (%) | | | | | | | |
| Never | 22(55.0%) | 10(26.3%) | 19(54.3%) | 0.017 | 0.950 | 0.015 | 0.010 |
| Ever/Current | 18(45.0%) | 28(73.7%) | 16(45.7%) | | | | |
| Alcohol status, n (%) | | | | | | | |
| Never | 12(30.0%) | 10(26.3%) | 11(31.4%) | 0.883 | | | |
| Current | 28(70.0%) | 28(73.7%) | 24(68.6%) | | | | |
| CKD, n (%) | 3(7.5%) | 2(5.3%) | 4(11.4%) | 0.618 | | | |
| Diabetes, n (%) | 10(25.0%) | 11(28.9%) | 8(22.9%) | 0.832 | | | |
| Tumor Clinical Stage | | | | | | | |
| I∼III | 25(62.5%) | 26(68.4%) | | 0.583 | | | |
| IV | 15(37.5%) | 12(31.6%) | | | | | |

PC: pancreatic cancer, BTC: Biliary-tract cancer, CKD: Chronic Kidney Disease

miRNAs were significantly up-regulated, and one miRNA was significantly downregulated in bile samples of PC and BTC compared with the control (P<0.05). Of these miRNAs, we selected six up-regulated miRNAs (miR-1275, miR-6891-5p, miR-3162-5p, miR-7107-5p, miR-4532, miR-3197) with high abundance and no significant difference in signal values within the PC and BTC cohorts for further analysis (S1 Fig and S1 Table).

## Quantitative analysis of bile miRNAs in patients with PC, BTC and control

Validation with quantitative PCR of these 6 up-regulated miRNAs was then performed in all bile samples. The Ct values for miR-16 were present in sufficient quantities and no significant differences (P > 0.05) in the 113 bile samples (S2 Fig), thus validating miR-16 as a reliable endogenous housekeeping. Of the six miRNAs, miR-3162-5p and miR-4532 were excluded from further analysis due to lower Ct values in the majority of individuals across all cohorts, and the remaining four miRNAs were finally validated in all bile samples: miR-1275, miR-6891-5p, miR-7107-5p and miR-3197. The primer sequences used are shown in S2 Table. As a result, the miR-1275 expression level in the bile of PC and BTC was significantly higher than that in the bile of the control (2-ΔΔCt = 1.57, p = 0.003 and 2-ΔΔCt = 0.55, p = 0.049, respectively). Although there was no significant difference in the miR-6891-5p expression level between BTC and the control, the miR-6891-5p expression level in PC was significantly higher

**Table 2. Laboratory characteristics.**

| | PC (n = 40) | BTC (n = 38) | Control (n = 35) | P value | PC vs Control | BTC vs Control | PC vs BTC |
|---|---|---|---|---|---|---|---|
| CA19-9 U/ml median (IQR) | 425(1,536) | 102(361) | 12(17.4) | <0.001 | <0.001 | <0.001 | 0.465 |
| T-Bil mg/dl median (IQR) | 7.5(6.3) | 5.4(12.3) | 0.8(1.0) | 0.001 | <0.001 | <0.001 | >0.999 |
| WBC, ×10³/μl median (IQR) | 5.3(2.9) | 6.0(2.6) | 5.9(5.0) | 0.187 | | | |
| TP,g/dl mean (s.d.) | 6.4(0.9) | 6.5(0.8) | 6.4(0.8) | 0.809 | | | |
| CRP,mg/dl median (IQR) | 0.5(1.3) | 0.7(2.8) | 0.1(1.6) | 0.024 | 0.213 | 0.021 | >0.999 |

PC: pancreatic cancer, BTC: Biliary tract cancer; CA19-9: carbohydrate antigen 19–9, T-Bil: total bilirubin, WBC: white blood cells, TP: total protein, CRP: C-reactive protein.

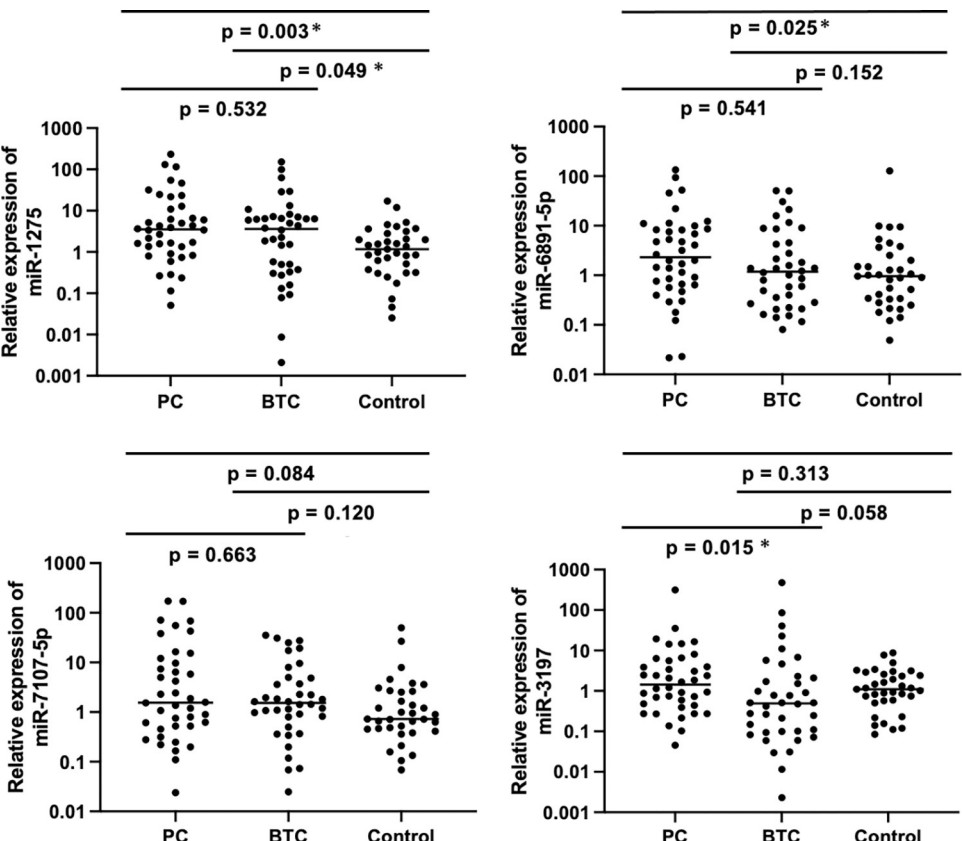

**Fig 1. The relative expression of four miRNAs in the bile of PC, BTC and controls (n = 113).** The horizontal line represents the median for each miRNA. miR-1275 in bile revealed significant differences between PC versus control (p = 0.003) and BTC versus control (p = 0.049). MiR-6891-5p in bile revealed significant differences compared PC versus control (p = 0.025). MiR-3197 in bile revealed significant differences between PC and BTC (p = 0.015). Other bile analyses showed no significant difference.

than that in the control (2-ΔΔCt = 3.28, p = 0.025). In addition, the miR-3197 expression level in the bile of PC was significantly higher than that in BTC (2-ΔΔCt = 0.56, p = 0.015). For miR-7107-5p expression, no differences between each cohort were found (Fig 1). In this study, we performed correlation analysis to evaluate the effect of jaundice on miRNAs because the malignant biliary stricture group tended to be in the setting of obstructive jaundice compared to the control; no significant relevant correlation between cholestasis in PC and BTC patients and the miRNA expression level was found (S3 Fig) The patient status of sex, smoking and clinical stage of PC and BTC were not affected (S4 Fig). Based on the results of quantitative PCR of these 6 miRNAs, the diagnostic value of quantification of miR-1275 and miR-6891-5p for PC and miR-1275 for BTC, combined with bile cytology, was finally evaluated.

## Diagnostic value of quantitative analysis of miR-1275 and miR-6891-5p combined with bile cytology to diagnose pancreatic cancer

Finally, the expression levels of only two miRNAs, miRNA-1275 and miRNA-6891-5p, were significantly higher than those in the control in PC. To verify the diagnostic value of the two miRNA expression levels in bile, ROC curve analysis was performed. The AUC values for miR-1275 and miR-6891-5p expression levels in bile to diagnose PC were 0.70 (95% confidence interval (CI) 0.58–0.82, p = 0.003) and 0.65 (95% CI 0.53–0.78, p = 0.025), respectively.

**Table 3. Diagnostic value of miR-1275, miR-6891-5p, bile cytology, miR-1275 combined with bile cytology and mi-6891-5p combined with bile cytology to detect biliary stricture with PC (PC: n = 40, control: n = 35).**

| | miR-1275 (%) | miR -6891-5p (%) | ERCP cytology (%) | miR1275 or Cytology (%) | miR6891-5p or Cytology (%) |
|---|---|---|---|---|---|
| Sensitivity (95% CI) | 70.0 (59.5–79.1) | 80.0 (70.4–88.2) | 70.0 (63.1–70.0) | 77.5 (70.7–77.5) | 77.5 (69.7–79.5) |
| Specificity (95% CI) | 60.0 (48.0–70.4) | 40.0 (29.0–49.4) | 100 (92.1–100) | 100 (92.2–100) | 97.1 (88.2–99.5) |
| PPV (95% CI) | 66.7 (56.7–75.4) | 60.4 (53.1–66.6) | 100 (90.1–100) | 100 (91.2–100) | 96.9 (87.1–99.4) |
| NPV (95% CI) | 63.6 (50.9–74.7) | 63.6 (46.1–78.6) | 74.5 (68.6–58.3) | 79.5 (73.4–79.5) | 79.1 (71.8–81.0) |
| Accuracy (95% CI) | 65.3 (54.1–75.1) | 61.3 (51.1–70.1) | 84.0 (76.6–84.0) | 88.0 (80.7–88.0) | 86.7 (78.3–88.9) |
| AUC (95% CI) | 0.70 (0.58–0.82) | 0.65 (0.53–0.78) | 0.85 (0.78–0.92) | 0.93* (0.87–0.99) | 0.89** (0.81–0.97) |

PPV: Positive predictive value, NPV: Negative predictive value, AUC: Area under the curve

*AUC value of quantitative analysis of miR-1275 in bile combined with bile cytology to diagnose biliary stricture with PC was significantly higher than cytology alone.

**There was no significant difference in comparison of quantitative analysis of miR-6891-5p combined with bile cytology with cytology alone.

The sensitivity, specificity, positive predictive value (PPV), negative predictive value (NPV) and diagnostic accuracy of the miR-1275 expression level in bile to diagnose PC were 70.0, 60.0, 66.7, 63.6 and 65.3, respectively; those of the miR-6891-5p expression level in bile to diagnose PC were 80.0, 40.0, 60.4, 63.6 and 61.3, respectively (Table 3). In this study, the sensitivity, specificity, PPV, NPV, accuracy and AUC of aspirated bile cytology alone to diagnose PC were 70.0, 100, 100, 74.5, 84.0 and 0.85, respectively. The sensitivity, specificity, PPV, NPV, accuracy and AUC to diagnose PC of miR-1275 expression level combined with bile cytology were 77.5, 100, 100, 79.5, 88.0 and 0.93, respectively; those of miR-6891-5p combined with bile cytology were 77.5, 97.1, 96.9, 79.1, 86.7 and 0.89, respectively (Table 3 and Fig 2). The AUC value of quantitative analysis of bile miR-1275 combined with bile cytology to diagnose PC was significantly higher than that of bile cytology alone (p = 0.014), although there was no significant difference in comparison of quantitative analysis of miR-6891-5p combined with bile cytology with cytology alone (p = 0.212).

## Diagnostic value of quantitative analysis of miR-1275 combined with bile cytology to diagnose biliary stricture with biliary tract cancer

Quantitative analysis showed that only the miR-1275 expression level in the bile of patients with BTC was significantly higher than that in the bile of controls. The AUC value for the miR-1275 expression level in bile to diagnose BTC was 0.63 (95% CI 0.50–0.77, p = 0.049). The sensitivity, specificity, PPV, NPV and diagnostic accuracy of the miR-1275 expression level in bile to diagnose BTC were 52.6, 80.0, 74.1, 60.9 and 63.0, respectively (Table 4). In this study, the sensitivity, specificity, PPV, NPV, accuracy and AUC of aspirated bile cytology alone to diagnose BTC were 65.8, 100, 100, 72.9, 82.2 and 0.83, respectively. The sensitivity, specificity, PPV, NPV, accuracy and AUC to diagnose BTC of miR-1275 expression level combined with bile cytology were 84.2, 94.3, 94.1, 84.6, 89.0 and 0.89, respectively (Table 4 and Fig 3). The miR-1275 expression level analysis did not show significant additional value for bile cytology in diagnosing BTC (p = 0.145).

## Disccusion

Biliary stricture is a frequently encountered condition in the clinical practice of pancreato-biliary areas and is caused by a variety of benign and malignant diseases. Although malignant biliary stricture can be caused by PC, BTC, hepatocellular carcinoma (HCC), neuroendocrine tumors, lymph node metastasis, and biliary tract metastasis from cancer of other organs, PC

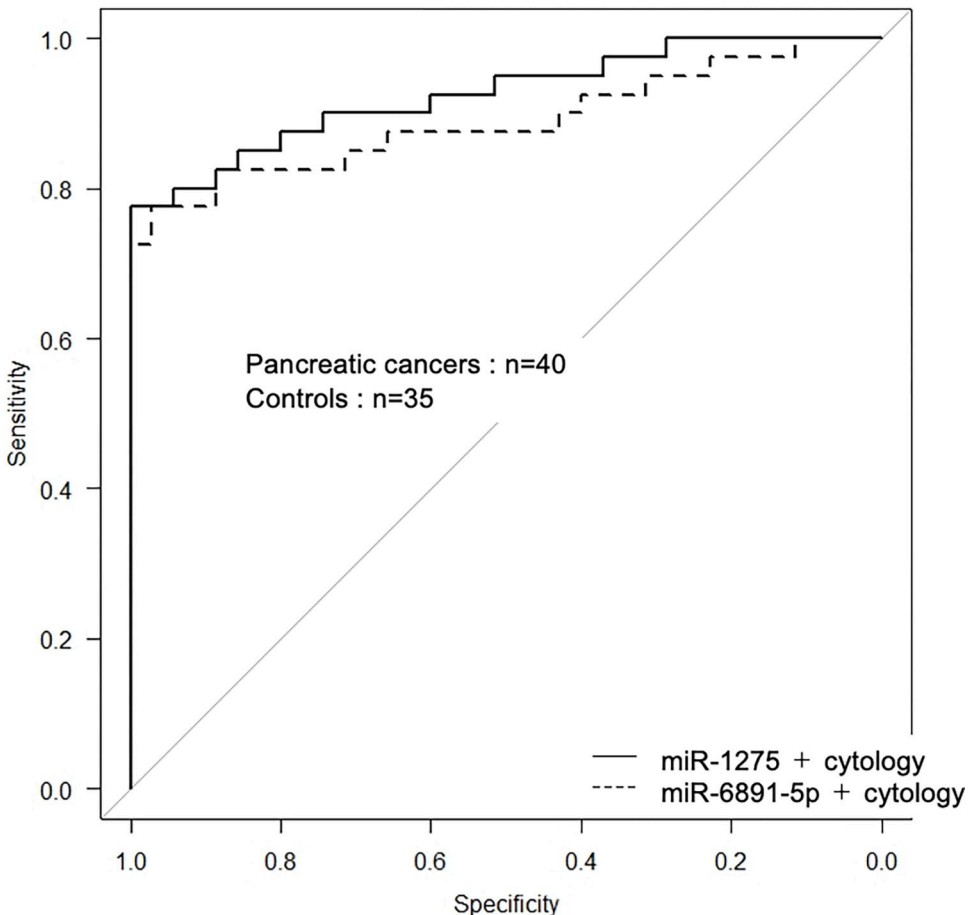

**Fig 2. Diagnostic performance of the combination of bile cytology and miRNA in bile for detecting PC.** Receiver operating characteristic curves illustrate the performance of the combination of bile cytology and miRNA quantification (miR-1275 and miR-6891-5p) in bile in distinguishing individuals with pancreatic cancer from those with benign biliary strictures.

and BTC account for the majority of causes. Surgery for these cancers is highly invasive, and a reliable preoperative diagnosis of benign or malignant disease is needed. Pathological diagnostic methods for malignant biliary stricture include bile cytology, brush cytology, endoscopic trans-papillary forceps biopsy and POCS-guided targeted biopsy. In a meta-analysis by

**Table 4. Diagnostic value of miR-1275, bile cytology and miR-1275 combined with bile cytology to detect biliary stricture with BTC (BTC: n = 38, control: n = 35).**

|  | miR-1275 (%) | Bile cytology (%) | Combination (%) |
|---|---|---|---|
| Sensitivity (95% CI) | 52.6 (37.3–67.5) | 65.8 (58.5–65.8) | 84.2 (75.7–87.9) |
| Specificity (95% CI) | 80.0 (64.1–90.0) | 100 (92.1–100) | 94.3 (85.1–98.3) |
| PPV (95% CI) | 74.1 (59.3–85.6) | 100 (88.9–100) | 94.1 (84.6–98.2) |
| NPV (95% CI) | 60.9 (52.2–67.6) | 72.9 (67.1–72.9) | 84.6 (76.3–88.2) |
| Accuracy (95% CI) | 65.8 (54.8–74.3) | 82.2 (74.6–82.2) | 89.0 (80.2–92.9) |
| AUC (95% CI) | 0.63 (0.50–0.77) | 0.83 (0.75–0.91) | 0.89 (0.80–0.97) * |

PPV: Positive predictive value, NPV: Negative predictive value, AUC: Area under the curve

*The combination of bile cytology and miR-1275 expression level analysis did not show significant additional value for diagnosing BTC.

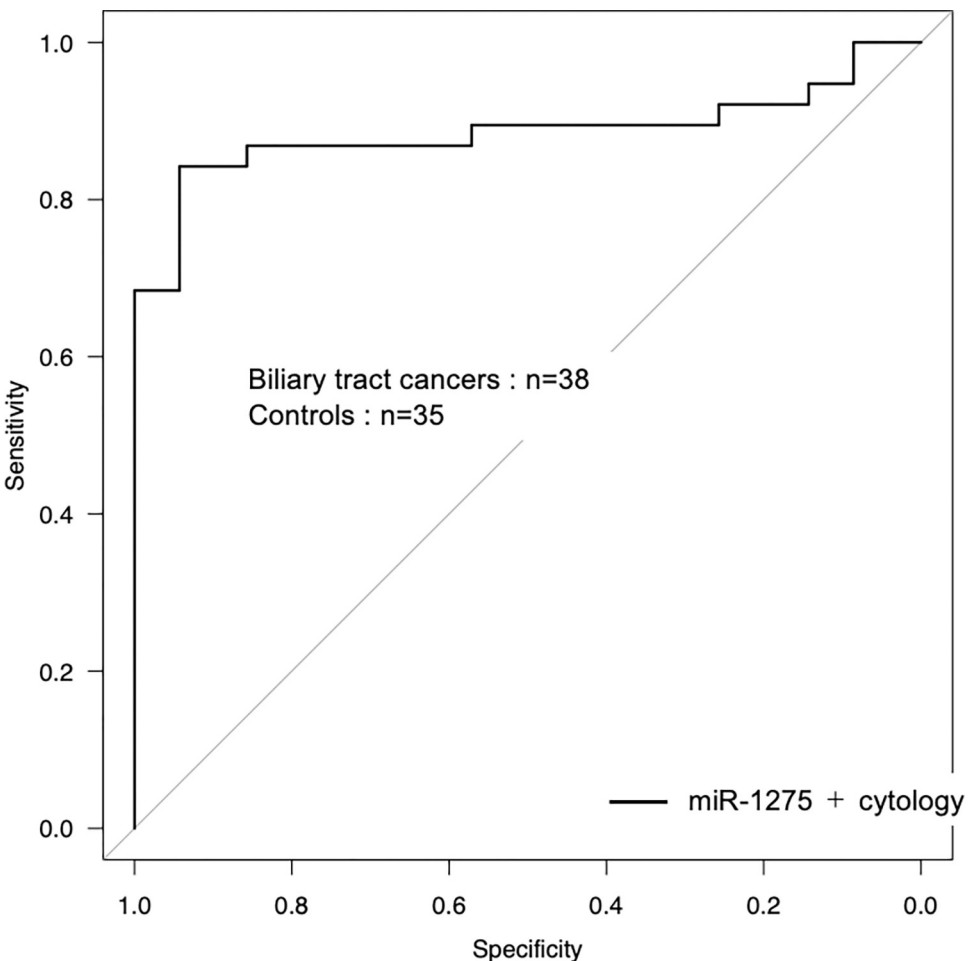

**Fig 3. Diagnostic performance of the combination of bile cytology and miRNA in bile for detecting BTC.** Receiver operating characteristic curves illustrate the performance of the combination of bile cytology and miRNA quantification (miR-1275) in bile in distinguishing individuals with biliary tract cancer from those with benign biliary strictures.

Navaneethan et al. [18], the sensitivity and specificity for the diagnosis of malignancy by endoscopic brush cytology were reported to be 45% and 99%, respectively, and those for trans-papillary forceps biopsy were 48% and 99%, respectively; their diagnostic sensitivity was not satisfactory. Many studies have reported that POCS-guided targeted biopsy has a higher sensitivity than a trans-papillary forceps biopsy and brushing cytology [19]. However, the specimens obtained by POCS-guided biopsy are usually small, and pathological diagnosis is often difficult. Recently, the usefulness of EUS-FNA for trans-papillary forceps biopsy-negative biliary stricture has been reported [20]. However, the risk of dissemination by EUS-FNA in hilar cholangiocarcinoma has also been reported [21]. Thus, the diagnosis of malignant biliary strictures is a clinical challenge and of special interest.

Recently, miRNAs have been reported to be applied as cancer biomarkers. A biomarker should be stable in fluids, accessible, disease-specific and easy to measure in the body. Many researchers have reported the stability of circulating miRNAs due to the formation of various complexes, unlike the majority of RNAs [22–24]. Therefore, miRNAs in serum and plasma have recently attracted attention as cancer biomarkers, and their usefulness has already been reported in various types of cancer, including gastric, colon, lung, breast, ovarian and prostate

cancers [25]. A variety of miRNAs have also been reported as biomarkers for PC and BTC, which are the major factors of malignant biliary stricture. Kim et al. examined a serum sample each from patients with PC, BTC and healthy controls and concluded that three serum miR-NAs, mir-744-5p, mir-409-3p, and mir-128-3p, were useful as potential biomarkers for PC and BTC diagnosis [26]. Moreover, Kojima et al. showed the value of a diagnostic index conducted by the combination of several predictive serum miRNAs in the detection of PC and BTC [27]. On the other hand, circulating miRNAs isolated from extracellular samples other than serum and plasma, such as saliva and urine, have also been reported to be equally useful for the diagnosis of PC and BTC [28, 29], and there have been few studies on the assessment of the miRNAs in bile to discriminate patients with BTC and PC from controls [10–14]. In particular, only one paper about miRNA in bile to detect PC could be found, which showed increased expression of miR-10b, miR-30c, miR-106b, miR-155, and miR-212 in plasma and bile appeared highly accurate in diagnosing PC [14].

Malignant biliary stricture is often accompanied by obstructive jaundice and cholangitis and requires biliary drainage procedures. Because bile samples are easily obtained during endoscopic drainage procedures, miRNAs in bile might be an efficient diagnostic cancer biomarker for malignant biliary stricture with a less invasive procedure and overcome the diagnostic limitation observed in conventional histocytological diagnosis methods. Bile is secreted by hepatocytes into the biliary canaliculi and directed through the biliary tract to the intestine, and bile is in direct contact with affected tissues by diseases causing biliary stricture and may be more likely to reflect pathological processes involving the biliary epithelium compared to serum and plasma [30]. For these reasons, bile may be an attractive sample to investigate molecules with diagnostic and prognostic potential for malignant biliary stricture. In this study, we evaluated the usefulness of quantification analysis of miRNAs in bile in the diagnosis of PC and BTC and whether it provides an additional effect to the diagnosis of PC and BTC when combined with bile cytology. Using highly sensitive microarrays to permit the simultaneous analysis of 2578 miRNAs that were recently updated in miRBase, we examined the expression profiles of comprehensive bile miRNAs in patients with PC, BTC and controls, and then validation was performed in 113 bile samples for the selected miRNAs. miR-1275 displayed higher levels in patients with PC and BTC than in those with benign biliary stricture. miR-6891-5p displayed higher levels in patients with PC than in those with benign biliary stricture. miR-3197 displayed higher levels in biliary stricture with PC than in biliary stricture with BTC. In particular, a combination of miR-1275 and aspirated bile cytology showed reliable diagnostic specificity and sensitivity in the diagnosis of biliary stricture with PC and a significantly greater additional diagnostic effect than bile cytology alone. Since PC tends to have poorer diagnostic ability for bile cytology than BTC, which is a tumor derived from the biliary epithelium [31], quantitative analysis of miRNAs in bile may be more useful in the diagnosis of biliary stricture caused by pancreatic cancer.

It has been widely reported that miR-1275 modulates the progression of various cancer types as a tumor-promoting or tumor-suppressing miRNA. In the literature, miR-1275 is upregulated in patients with nonsmall lung cancer, squamous carcinoma of the head and neck and chronic myelogenous leukemia [32–34], whereas other studies have reported that this miRNA is downregulated in gastric and nasopharyngeal carcinoma and functions as a tumor suppressor [35–37]. To date, there is only one report of quantitative analysis of miR-1275 in PC. Lei et al. analyzed the expression levels of miR-1275 in PC tissues and normal pancreatic tissues of 20 PC cases and found a significant decrease in miR-1275 in PC tissues in comparison with matched noncancerous tissues [38]. Furthermore, they reported that overexpression of miR-1275 in PC cell lines significantly inhibited cell proliferation, metastasis, and invasion. Thus, our study showed the opposite results to Lei's study, and these results might depend on

the differences in the target samples for miRNA isolation: tissue and bile. There are some studies with the contradictory results for the same type of miRNA depending on the type of sample used. Flammang et al. showed a downregulation of tissue-derived miR-192-5p and an upregulation of blood miR-192-5p in patients with pancreatic ductal adenocarcinoma compared to healthy controls [39]. They state that the reason may involve not only benign and malignant factors, but also multifactorial involvement, such as the presence of inflammation. Including this study, there are only two reports of quantitative analysis of miR-1275 in PC. Therefore, a certain trend may be seen as further studies are accumulated. In addition, there have been no reports of an association between miR-1275 and BTC. Although miR-6891-5p was up-regulated in patients with selective IgA deficiency [40] and downregulated in coronary artery disease [41], there have been no reports of miR-6891-5P as a biomarker for malignant tumors. The mechanism of miR-1275 and miR-6891-5p dysregulation in patients with PC and BTC remains unclear but may involve the differences in the sample acquisition, processing, storage, source type and reference gene. We used the miRWalk online database (http://zmf.umm.uni-heidelberg.de/apps/zmf/mirwalk2/index.html) [42] and Kyoto Encyclopedia of Genes and Genomes (KEGG) pathway to predict dysregulated regulatory genes and signaling pathways of miR-1275 and miR-6891-5p. The results indicated that numerous target genes of miR-1275 are associated with the mitogen-activated protein kinase (MAPK) signaling pathway and miR-6891-5p is associated with the phosphatidylinositol-3 kinase (PI3K)-Akt signaling pathway. The MAPK signaling pathway and PI3K-Akt signaling pathway are downstream signals of *KRAS*, and mutations have been reported to be involved in the onset of PC and BTC [43–45], supporting the high expression of miR-1275 and miR-6891-5p in PC and BTC in this study.

Several reports on quantitative analysis of miRNAs in bile have isolated miRNAs from whole bile, while others have analyzed miRNAs contained in exosomes, which are referred to as extracellular vesicles (EVs), in a broad sense in bile [10–14]. In this study, we used whole bile as a source for miRNA profiling. An exosome is a membrane vesicle approximately 30 to 120 nm in size secreted from various cells. Proteins, lipids, nucleic acids, and so on are contained within exosomes or on their membranes. Exosomes are taken up by cells and are involved in intercellular communication [46]. Recent publications described that the miRNA species from free-floating cells are unstable and thus not suitable for diagnostic approaches, in contrast to the miRNAs derived from exosomes, while some publications suggest that a significant additional amount of extracellular miRNAs are not present in exosomes [10, 47, 48]. Shigehara et al. conducted fractionation of bile samples by differential centrifugation and compared the expression level of the miRNAs in each fraction. They concluded that most of the bile miRNAs were not found in small vesicles but in whole cells and nuclei. Consequently, the detailed mechanisms by which the circulating miRNAs reach the circulation in bile are not fully understood, and the miRNA profiles for each fraction in bile are controversial. However, the present study used whole bile samples to make it more clinically relevant and could establish a quantitative analysis for the miRNAs in whole bile, which could provide promising results for the reliable diagnosis of malignant biliary strictures when combined with cytology. Further analysis is needed to validate the miRNA profiles for each fraction in bile.

There are other limitations in this study. First, external validity was limited because all participants were recruited at the Nihon University Itabashi Hospital. Second, bile sampling with the invasive nature of biliary drainage also makes it difficult to obtain samples from completely healthy individuals.

In conclusion, our results herein document that miR-1275 and miR-6891-5p were significantly up-regulated in biliary strictures with PC and BTC compared to benign biliary strictures and that the combination of aspirated bile cytology plus miR-1275 quantification in bile can discriminate biliary strictures with PC from benign individuals with fairly good sensitivity and

specificity. The identified dysregulated miRNAs might be a potential biomarker and therapeutic target but have to be validated in larger cohorts, as well as to assess their usefulness as markers that correlate with survival and response to therapy for individuals with these cancers.

## Supporting information

**S1 Checklist. STROBE statement—checklist of items that should be included in reports of observational studies.**
(DOCX)

**S1 Fig. Heatmap representing miRNA expression in bile samples from 9 cases (3 pancreatic cancers, 3 biliary tract cancers, 3 controls).** Heatmap representing miRNA expression in bile samples from 9 cases (3 pancreatic cancers, 3 biliary tract cancers, 3 controls). Thirty-five of 2578 miRNAs were significantly up-regulated, and one miRNA was significantly downregulated in PC and BTC bile samples compared with the control (P<0.05).
(TIF)

**S2 Fig. The Ct values of miR-16 in the bile of PC, BTC and controls (n = 113).** The Ct values for miR-16 were present in sufficient quantities and no significant differences (P > 0.05) in the 113 bile samples, thus validating miR-16 as a reliable endogenous housekeeping.
(TIF)

**S3 Fig. Correlation between the miRNA measurement and cholestasis analyzed by linear regression test (n = 113).** Correlation between the miRNA measurement and cholestasis analyzed by linear regression test (n = 113). No significant correlation was observed between the quantification of the miRNAs in bile and total bilirubin.
(TIF)

**S4 Fig. Differential expression of miRNA in bile is illustrated as a function of sex, smoking status (ever/current or never) and tumor stage.** Differential expression of miRNA in bile is illustrated as a function of sex, smoking status (ever/current or never) and tumor stage. No significant differences in any miRNA expression levels were found in these clinical statuses.
(TIF)

**S1 Table. Selected miRNA over-expressed in sets of pooled bile from patients with pancreatic cancer and bile tract cancer compared to those from healthy controls.**
(DOCX)

**S2 Table. Sequence of primers for quantitative PCR.**
(DOCX)

## Author Contributions

**Conceptualization:** Noriyuki Kuniyoshi, Hiroo Imazu, Motomi Yamazaki.

**Data curation:** Noriyuki Kuniyoshi, Motomi Yamazaki.

**Formal analysis:** Noriyuki Kuniyoshi.

**Investigation:** Noriyuki Kuniyoshi, Motomi Yamazaki.

**Methodology:** Hiroo Imazu.

**Project administration:** Hiroo Imazu.

**Resources:** Noriyuki Kuniyoshi, Motomi Yamazaki, Suguru Hamana, Shuzo Nomura, Jo Hayama, Rota Osawa, Koji Yamada, Mariko Fujisawa, Kei Saito.

**Supervision:** Hirofumi Kogure.

**Validation:** Noriyuki Kuniyoshi.

**Writing – original draft:** Noriyuki Kuniyoshi.

**Writing – review & editing:** Hiroo Imazu, Ryota Masuzaki, Hirofumi Kogure.

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
