## [Decision Letter · Decision Letter 0]

11 May 2023

PONE-D-23-07350Diagnostic utility of quantitative analysis of microRNA in bile samples obtained during endoscopic retrograde cholangiopancreatography for malignant biliary stricturesPLOS ONE

Dear Dr. Imazu,

Thank you for submitting your manuscript to PLOS ONE. After careful consideration, we feel that it has merit but does not fully meet PLOS ONE’s publication criteria as it currently stands. Therefore, we invite you to submit a revised version of the manuscript that addresses the points raised during the review process.

We look forward to receiving your revised manuscript.

Kind regards,

Jincheng Wang

Academic Editor

PLOS ONE

Additional Editor Comments:

This paper is well written and the data are well presented. I think this paper quality can be improved if authors can provide more evidence such as bioinformatics analysis from TCGA.

Reviewers' comments:

Reviewer's Responses to Questions

**Comments to the Author**

1. Is the manuscript technically sound, and do the data support the conclusions?

Reviewer #1: Partly

2. Has the statistical analysis been performed appropriately and rigorously? 

Reviewer #1: Yes

3. Have the authors made all data underlying the findings in their manuscript fully available?

Reviewer #1: Yes

4. Is the manuscript presented in an intelligible fashion and written in standard English?

Reviewer #1: Yes

5. Review Comments to the Author

Reviewer #1: Kuniyoshi et al. evaluated the level of a large set of miRNAs in bile samples from PC (n=3), BTC (n=3) and control (n=3) patients and after qPCR validation of 6 miRNAs, they highlighted 4 significantly up-regulated. Finally, they suggest that the levels of miR-1275 and miR-6891-5p in bile may help in the diagnosis of PC and BTC after ERCP.

The manuscript is well written, with a considerable cohort of samples and the aim well defined. Moreover, although they are cautious with the conclusions, the utility is far from the clinical use.

I have some comments:

- Although significant, the differences between tumor patients and the controls is low talking about Ct. Which would be the threshold to consider that a blind bile sample analyzed has the miRNA high or low and therefore would be consider as malignant or benign?

- Although in Material&Methods it is described (page 7, lines 96-98), I am not sure I understood it correctly. Patients included in the study were diagnosed with a malignancy in the samples collected during the ERCP. According to Fig 1 there are control samples with high levels of miRNAs (comparable to tumor patients). Have you follow these patients to know whether they were later diagnosed with a tumor? This would really improve the utility you would anticipate the diagnosis.

- Authors have used miR-16 as housekeeping for the qPCR analysis and all the conclusions are based on those results. Although they justify that other authors used miR-16 as housekeeping, the other studies were done in plasma or tissue samples. Authors should include a supplementary figure reporting the miR-16 value for the 113 samples analysed to demonstrate that it is a good housekeeping.

- I found contradictory that in page 25- lines 380-383 authors said that there is one 1 paper about miRNA in bile for PC diagnosis (the ref 14), and then in page 29- lines448-451 they said that there are several reports on qPCR analysis of miRNAs in bile. The difference is the type of tumor? I mean that in these “several reports” they do not have PC patients? Or what? Moreover, references for these statement in page 29 are missing.

- I also find weird that contradictory results are found between this study and Lei et al in terms of miR-1275 levels in PC patients (REF 38). Of course as authors discussed I agree with the difference in the sample used (tissue vs bile), but normally there should not be these discrepancy. I think it would be important to analyzed some (n=10-15) pair samples (tissue-bile) in your hands.

- To perform the miRNA expression analysis 130 ng of total RNA from each sample was used. Does this mean that 45 ng of each of the 3 samples were pooled?

- Titles in Results section should be more informative than a simple “Patients” or “Quantitative PCR”.

6. PLOS authors have the option to publish the peer review history of their article (what does this mean?). If published, this will include your full peer review and any attached files.

Reviewer #1: No

---

## [Author Response · Author response to Decision Letter 0]

9 Jun 2023

We appreciate the time and effort each of the edditers and reviewers have dedicated to providing insightful feedback on ways to strengthen our paper. Thus, it is with great pleasure that we resubmit our article for further consideration. We have incorporated changes that reflect the detailed suggestions you have graciously provided. We also hope that our edits and the responses we provide satisfactorily address all the issues and concerns the reviewers have noted.

---

## [Editor Report · Decision Letter 1]

21 Jul 2023

Diagnostic utility of quantitative analysis of microRNA in bile samples obtained during endoscopic retrograde cholangiopancreatography for malignant biliary strictures

PONE-D-23-07350R1

Dear Dr. Imazu,

We’re pleased to inform you that your manuscript has been judged scientifically suitable for publication and will be formally accepted for publication once it meets all outstanding technical requirements.

Kind regards,

Gopal Krishna Dhali, MBBS, MD, DM

Academic Editor

PLOS ONE

Additional Editor Comments (optional):

Thank you for the amendments.

---

## [Editor Report · Acceptance letter]

31 Jul 2023

PONE-D-23-07350R1 

Diagnostic utility of quantitative analysis of microRNA in bile samples obtained during endoscopic retrograde cholangiopancreatography for malignant biliary strictures 

Dear Dr. Imazu:

I'm pleased to inform you that your manuscript has been deemed suitable for publication in PLOS ONE. Congratulations! Your manuscript is now with our production department. 

Kind regards, 

on behalf of

Dr. Gopal Krishna Dhali 

Academic Editor

PLOS ONE